# Seven New Species of Eurotiales (Ascomycota) Isolated from Tidal Flat Sediments in China

**DOI:** 10.3390/jof9100960

**Published:** 2023-09-25

**Authors:** Chang Liu, Xin-Cun Wang, Zhi-He Yu, Wen-Ying Zhuang, Zhao-Qing Zeng

**Affiliations:** 1State Key Laboratory of Mycology, Institute of Microbiology, Chinese Academy of Sciences, Beijing 100101, China; ymliu19@163.com (C.L.); wangxc@im.ac.cn (X.-C.W.); zhuangwy@im.ac.cn (W.-Y.Z.); 2College of Life Sciences, Yangtze University, Jingzhou 434025, China; zhiheyu@hotmail.com

**Keywords:** *Aspergillus*, biodiversity, filamentous fungi, *Penicillium*, phylogeny, *Talaromyces*, taxonomy

## Abstract

Tidal flats have been reported to contain many microorganisms and play a critical role in maintaining biodiversity. In surveys of filamentous fungi from tidal flat sediments in China, seven new species of Eurotiales were discovered and described. Morphological characteristics and DNA sequence analyses of combined datasets of the *BenA*, *CaM*, and *RPB2* regions support their placements and recognition as new species. *Aspergillus liaoningensis* sp. nov. and *A. plumeriae* sp. nov. belong to sections *Candidi* and *Flavipedes* of subgenus *Circumdati*, and *A. subinflatus* sp. nov. is a member of section *Cremei* of subgenus *Cremei*. *Penicillium danzhouense* sp. nov., *P. tenue* sp. nov., and *P. zhanjiangense* sp. nov. are attributed to sections *Exilicaulis* and *Lanata-Divaricata* of subgenus *Aspergilloides*. *Talaromyces virens* sp. nov. is in section *Talaromyces*. Detailed descriptions and illustrations of these novel taxa are provided. Their differences from close relatives were compared and discussed.

## 1. Introduction

Tidal flats, which link the ocean and the land, contain plentiful microorganisms [1]. Filamentous fungi were reported to be dominant in the intertidal fungal ecosystem [2,3]. The order Eurotiales is one of the most abundant groups, which contains five families with about 28 genera, including *Aspergillus* P. Micheli, *Penicillium* Link, and *Talaromyces* C.R. Benj. [4]. These genera are economically important in the fields of human health, agriculture, industry, and pharmaceutics [2,4,5,6,7]. For example, *A. fumigatus* Fresen. and *T. marneffei* (Segretain, Capponi & Sureau) Samson, N. Yilmaz, Frisvad & Seifert are two well-known human pathogens [8,9]. Penicillin, an effective anti-infective drug [10], was produced by *P. chrysogenum* Thom. *Aspergillus oryzae* (Ahlb.) Cohn can be used in food fermentation and was reported as a producer of enzymes [11,12]. Therefore, the discovery of these fungi is of theoretical and practical importance.

Due to the sophisticated classification, infrageneric taxonomy has commonly been used for *Aspergillus*, *Penicillium*, and *Talaromyces* [4]. With the application of multiple loci phylogeny, the inter-specific relationships within these genera have become more clear [13,14,15]. Currently, the genus *Aspergillus* comprises 483 species belonging to 27 sections [11,16,17,18,19,20,21,22], *Penicillium* contains 530 species in 33 sections [20,22,23,24,25,26,27,28,29,30,31,32,33,34,35,36,37,38], and *Talaromyces* includes 204 species in 7 sections [6,18,32,39,40,41,42]. They show a broad range of habitats, such as woody substratum, sandy soil, tidal flats, water, and indoor air [43,44,45,46,47].

During the examinations of filamentous fungi isolated from tidal flat sediments in different provinces in China, seven undescribed taxa were encountered. Judging by the cultural and microscopic characteristics, they belong to *Aspergillus*, *Penicillium*, and *Talaromyces*. Their taxonomic placements were further confirmed by carrying out multilocus phylogenetic analyses of β-tubulin (*BenA*), calmodulin (*CaM*), and the second-largest subunit of RNA polymerase II (*RPB2*). The distinctions between the novel species and their close relatives were compared.

## 2. Materials and Methods

### 2.1. Sampling and Fungal Isolation

Tidal flat sediment samples were collected from Guangdong, Hainan, and Liaoning provinces from August to October 2020. Sediment samples were kept at 4 °C until used. Strains were isolated using a 3% sea salt medium with the dilution plate method and were preserved in the China General Microbiological Culture Collection Center (CGMCC). Dried cultures were deposited in the Herbarium Mycologicum Academiae Sinicae (HMAS).

### 2.2. Morphological Observations

Colony characteristics were observed and described following the method of Visagie et al. [14]. Four standard growth media were used: Czapek yeast autolysate agar (CYA, yeast extract Oxoid), malt extract agar (MEA, Amresco), yeast extract agar (YES), and potato dextrose agar (PDA) [48,49,50]. A twenty-five percent lactic acid (LA) solution was used as the mounting medium for the microscopic examinations of structures and measurements of the conidial head, stipe, phialide, vesicle, and conidia. The methods for inoculation, morphological observation, and digital recordings were performed following previous studies [51,52].

### 2.3. DNA Extraction, PCR Amplification, and Sequencing

Genomic DNA was extracted from fungal mycelium with the Plant Genomic DNA Kit (Tiangen Biotech (Beijing) Co., Ltd., Beijing, China). The sequences of nuclear the ribosomal DNA ITS1-5.8S-ITS2 (ITS), *BenA*, *CaM*, and *RPB2* regions were amplified on an ABI 2720 Thermal Cycler (Applied Biosciences, Foster City, CA, USA) with the primer pairs ITS5 and ITS4 [53], T1 and Bt2a (or Bt2b) [54,55], CMD5 and CMD6 [56], and fRPB2-5F and fRPB2-7cR [57], respectively. PCR products were sequenced in both directions on an ABI 3730 DNA Sequencer (Applied Biosciences, Foster City, CA, USA).

### 2.4. Phylogenetic Analyses

The newly generated sequences and those retrieved from GenBank are listed in Table 1. They were assembled and aligned with BioEdit 7.0.5 [58] and manually edited. To evaluate statistical congruence amongst the loci *BenA*, *CaM*, and *RPB2*, the partition homogeneity test (PHT) was performed in PAUP*4.0b10 [59] with 1000 replicates. To determine the positions of these strains, the datasets of these regions belonging to *Aspergillus* sect. *Candidi*, *Cremei* and *Flavipedes*, *Penicillium* sect. *Exilicaulis* and *Lanata-Divaricata*, and *Talaromyces* sect. *Talaromyces* were compiled and analyzed by the maximum likelihood (ML) and Bayesian inference (BI) methods. ML analysis was performed with the default GTRCAT model using RAxML [60]. The BI analysis was conducted by MrBayes 3.2.5 [61]. Nucleotide substitution models were determined by MrModeltest 2.3 [62]. Dendrogram trees were visualized and edited using TreeView v. 1.6.6 [63] and FigTree v. 1.4.4 (http://tree.bio.ed.ac.uk/software/figtree/ (accessed on 25 November 2018)). A Bayesian inference posterior probability (BIPP) greater than 90% and a maximum likelihood bootstrap support (MLBS) greater than 70% were shown at the nodes.

## 3. Results

### 3.1. Phylogenetic Analyses

To determine the positions of the *Aspergillus* strains, *Hamigera avellanea* Stolk & Samson and *Penicillium expansum* Link were used as outgroup taxa. The partition homogeneity tests (*p* = 0.01 and 0.25, respectively) indicated that the individual partitions were not highly incongruent [64]; thus, these three loci were combined for the phylogenetic analyses. The phylogenetic trees showed that strains CGMCC 3.25201 and 3.25202 were located in sect. *Candidi* and *Flavipedes*, respectively (Figure 1). The strain CGMCC 3.25201 shared a close relationship with *A. subalbidus* Visagie, Hirooka & Samson (BIPP/MLBS = 91%/89%), while CGMCC 3.25202 clustered with *A. movilensis* A. Nováková, Hubka, S.W. Peterson & M. Kolařík, receiving high supporting values (BIPP/MLBS = 100%/100%). The strain CGMCC 3.25203 grouped with other members of sect. *Cremei* ser. *Inflati*, receiving high statistic values (BIPP/MLBS = 100%/100%) (Figure 2).

In the phylogenetic analyses of *Penicillium* sect. *Exilicaulis*, *H. avellanea* and *A. glaucus* (L.) Link served as outgroup taxa. The partition homogeneity test (*p* = 0.01) indicated that the individual partitions were not highly incongruent [64]; thus, these three loci were combined for the phylogenetic analyses. The phylogenetic trees showed that strains CGMCC 3.25204 and 3.25205 were located in ser. *Erubescentia* (BIPP/MLBS = 100%/93%) (Figure 3). The strain CGMCC 3.25204 was clustered with *P. canis* S.W. Peterson, while strain CGMCC 3.25205 was grouped with *P. striatisporum* Stolk, both receiving full support.

In the phylogenetic analyses of *Penicillium* sect. *Lanata-Divaricata*, *H. avellanea* and *P. glabrum* (Wehmer) Westling were used as outgroup taxa. The partition homogeneity test (*p* = 0.01) indicated that the individual partitions were not highly incongruent [64]; thus, these three loci were combined for the phylogenetic analyses. The strain CGMCC 3.25206 was placed in ser. *Janthinella* and clustered with *P. janthinellum* Biourge with high supporting values (BIPP/MLBS = 100%/100%) (Figure 4).

To determine the position of the *Talaromyces* strain, *T. trachyspermus* (Shear) Stolk & Samson and *T. chongqingensis* X.C. Wang & W.Y. Zhuang were used as outgroup taxa. The partition homogeneity test (*p* = 0.01) indicated that the individual partitions were not highly incongruent [64]; thus, these three loci were combined for the phylogenetic analyses. The phylogenetic tree showed that strain CGMCC 3.25207 was grouped with the others of sect. *Talaromyces* (BIPP/MLBS = 100%/100%) and was closely related to *T. xishaensis* X.C. Wang, L. Wang & W.Y. Zhuang (BIPP/MLBS = 100%/98%) (Figure 5).

### 3.2. Taxonomy

***Aspergillus* *liaoningensis*** C. Liu, Z.Q. Zeng & W.Y. Zhuang, sp. nov. Figure 6.

**Fungal Names**: FN 571613.

**Etymology:** The specific epithet refers to the type locality “Liaoning Province” of the fungus.

In: *Aspergillus* subgen. *Circumdati* sect. *Candidi* ser. *Candidi*.

**Typification:** CHINA, Liaoning Province, Donggang City, Zhongshan District, Yalu River Wetland Park, 39°49′0″ N 124°3′20″ E, in fluvial sediments, 13 October 2020, Chang Liu, tt32414 (holotype HMAS 247877, ex-type strain CGMCC 3.25201).

**DNA barcodes:** ITS ON563148, *BenA* ON231293, *CaM* ON470836, *RPB2* ON470844.

**Colony diam.:** 7 days, 25 °C (unless stated otherwise): CYA 20–23 mm; CYA 37 °C, 8–9 mm; CYA 5 °C no growth; MEA 13–14 mm; PDA 16–22 mm; YES 18–20 mm.

**Colony characteristics:** On CYA 25 °C, 7 days: Colonies irregular, plain, cracked; margins narrow, nearly entire; mycelia white to cream; texture velutinous to floccose; sporulation dense; conidia en masse cream; no exudate, no soluble pigments; reverse cream to light yellow, white at periphery. On CYA 37 °C, 7 days: Colonies irregular, slightly protuberant in centers; margins narrow to moderately wide; mycelia cream; texture velutinous to floccose; sporulation dense; conidia en masse white to cream; no exudate, no soluble pigments; reverse dark cream, brown at centers. On MEA 25 °C, 7 days: Colonies nearly circular, slightly protuberant in centers; margins narrow to moderately wide, nearly entire; mycelia white to light cream; texture velutinous to floccose; sporulation dense in center, light cream; no exudate, no soluble pigments; reverse cream, brown at centers, white at periphery. On PDA 25 °C, 7 days: Colonies irregular, plain, cracked; margins narrow, nearly entire; mycelia white to grey; texture velutinous to floccose; sporulation dense; conidia en masse grey; no exudate, no soluble pigments; reverse cream, brown at centers, white at periphery. On YES 25 °C, 7 days: Colonies nearly circular, protuberant in centers, edges irregular; margins narrow, nearly entire; mycelium light grey at center, white at margin; texture velutinous to floccose; sporulation moderately dense; no exudate, no soluble pigments; reverse white, light yellow at centers.

**Micromorphology:** Conidial heads radiate; stipes thick walls, smooth, hyaline, not septate, 50–260 × 5.0–7.5 µm; vesicles globose to broad ellipsoidal, 5.5–17.3 × 5.3–17.1 µm; biseriate; metulae cylindrical to obovate, 5.2–8.0 × 3.3–5.1 µm, covering two-thirds to almost the entire surface of the vesicle; phialides flask-shaped to acerose, 4.8–8.4 × 2.3–3.1 µm; conidia globose to subglobose, smooth, 2.6–4.2 µm in diam.

**Note:** This species is phylogenetically related to *A. subalbidus* (Figure 1), but the latter differs in that it lacks growth on CYA at 37 °C, has faster growth rates on MEA (17–19 mm) and YES (25–30 mm) at 25 °C, and its colonies do not crack on CYA and PDA [14].

***Aspergillus plumeriae*** C. Liu, Z.Q. Zeng & W.Y. Zhuang, sp. nov. Figure 7.

**Fungal Names:** FN 571614.

**Etymology:** The specific epithet refers to the yellowish-white colony on PDA.

In: *Aspergillus* subgen. *Circumdati* sect. *Flavipedes* ser. *Spelaei*.

**Typification:** CHINA, Liaoning Province, Dalian City, Zhongshan District, Binhai East Road, 38°52′1″ N 121°41′20″ E, in tidal flat sediments, 12 October 2020, Chang Liu, tt30226 (holotype HMAS 247878, ex-type strain CGMCC 3.25202).

**DNA barcodes:** ITS ON563147, *BenA* ON231292, *CaM* ON470835, *RPB2* ON470843.

**Colony diameter:** 7 days, 25 °C (unless stated otherwise): CYA 24–26 mm; CYA 37 °C no growth; CYA 5 °C no growth; MEA 18–22 mm; PDA 17–19 mm; YES 25–26 mm.

**Colony characteristics:** On CYA 25 °C, 7 days: Colonies nearly circular, plain, slightly protuberant in centers; margins narrow to moderately wide, nearly entire; mycelia white to cream; texture velutinous; sporulation dense, conidia en masse white; no exudate, no soluble pigments; reverse light yellow. On MEA 25 °C, 7 days: Colonies nearly circular, wrinkled, slightly protuberant in centers, radially sulcate; margins moderately wide, nearly entire; mycelia white to cream; texture velutinous; sporulation dense in center, cream to light yellow; no exudate, no soluble pigments; reverse yellowish-brown. On PDA 25 °C, 7 days: Colonies nearly circular, wrinkled, slightly concave at centers, radially sulcate; margins moderately wide, nearly entire; mycelia white, bright yellow at center; texture velutinous to floccose; sporulation dense in center, bright yellow; no exudate, no soluble pigments; reverse light–brown at centers, white at periphery. On YES 25 °C, 7 days: Colonies nearly circular, wrinkled, slightly concave at centers, radially sulcate; margins moderately wide, nearly entire; mycelium white to cream at center, white at margin; texture velutinous; sporulation moderately dense, white to cream; no exudate, no soluble pigments; reverse white to light yellow.

**Micromorphology:** Conidial heads radiate; stipes thick walls, smooth, hyaline or blackish, not septate, longer than 340 µm; vesicles globose to subglobose, 13–22.7 µm in diam.; biseriate; metulae cylindrical, 4.5–7.8 × 2.6–4.9 µm, covering two-thirds to almost the entire surface of the vesicle; phialides flask-shaped to acerose, slightly curved at the mouth, 5.4–8.4 × 1.7–2.9 µm; conidia globose to subglobose, smooth, 2.6–3.3 µm in diam.

**Note:** Among the known species of *Aspergillus*, *A. plumeriae* is distinct because of its yellowish-white colony on PDA. It is phylogenetically related to *A. movilensis* (Figure 1), but the latter differs in its ability to grow at 37 °C and production of pyriform vesicles with smaller sizes (5.0–16 µm in diam.) [65].

***Aspergillus subinflatus*** C. Liu, Z.Q. Zeng & W.Y. Zhuang, sp. nov. Figure 8.

**Fungal Names:** FN 571615.

**Etymology:** The specific epithet refers to the similarity of the fungus to *A. inflatus*.

In: *Aspergillus* subgen. *Cremei* sect. *Cremei*. Ser. *Inflati.*

**Typification:** CHINA, Hainan Province, Ledong Li Autonomous County, Liguo Town, 108°56′22″ N 18°24′38″ E, in mangrove sediments, 3 September 2020, Hai-Yan Zhu, tt14122 (holotype HMAS 247879, ex-type strain CGMCC 3.25203).

**DNA barcodes:** ITS ON563146, *BenA* ON231291, *CaM* ON470834, *RPB2* ON470845.

**Colony diameter:** 7 days, 25 °C (unless stated otherwise): CYA 16–17 mm; CYA 37 °C 4–5 mm; CYA 5 °C no growth; MEA 13–14 mm; PDA 17–18 mm; YES 14–16 mm.

**Colony characteristics:** On CYA 25 °C, 7 days: Colonies nearly circular, wrinkled, protuberant in centers, radially sulcate; margins narrow to moderately wide, nearly entire; mycelia white to grey; texture velutinous; sporulation dense; conidia en masse grey; no exudate, no soluble pigments; reverse light yellow. On CYA 37 °C, 7 days: Colonies nearly circular, lain; mycelia light pink to white; texture velutinous; no sporulation; no exudate, no soluble pigments; reverse light pink to white. On MEA 25 °C, 7 days: Colonies nearly circular, slightly wrinkled, slightly protuberant in centers; margins wide to moderately wide, nearly entire; mycelia white to grey; texture velutinous to floccose; sporulation dense in center, grey; no exudate, no soluble pigments; reverse yellowish-brown, yellow to white at centers. On PDA 25 °C, 7 days: Colonies nearly circular, slightly wrinkled, slightly protuberant in centers; margins moderately wide, nearly entire; mycelia white to light grey; texture velutinous to floccose; sporulation dense in center, light grey; no exudate, no soluble pigments; reverse pale yellow. On YES 25 °C, 7 days: Colonies nearly circular, wrinkled, protuberant in centers; margins narrow, nearly entire; mycelium white, light grey at center; texture velutinous to floccose; sporulation moderately dense, white; no exudate, no soluble pigments; reverse yellow, white at periphery.

**Micromorphology:** Conidial heads radiate; stipes smooth-walled, slightly swollen at the apex, 12–318 × 3.5–5.0 µm; vesicle ellipsoidal, 5.3–17 × 3.1–6.4 μm; metulae developing successively on the vesicle and also occurring on its subterminal and terminal portion, swollen, cylindrical to obovate, 3.8–9.4 × 2.5–4.1 µm; phialides flask-shaped to acerose, tapering into thin neck, 2.5–7.9 × 1.9–2.6 µm; conidia globose to subglobose, smooth to finely rough, 1.6–2.3 μm in diam.

**Note:** This species is morphologically similar and phylogenetically related to *A. inflatus* (Figure 2) but differs in that its metulae occur at the subterminal and terminal positions [13,66]. Moreover, there are 69 bp, 74 bp, and 88 bp divergences in the *BenA*, *CaM*, and *RPB2* regions between the ex-type cultures of the two taxa (CGMCC 3.25203 and CBS 682.70).

***Penicillium danzhouense*** C. Liu, Z.Q. Zeng & W.Y. Zhuang, sp. nov. Figure 9.

**Fungal Names:** FN 571616.

**Etymology:** The specific epithet refers to the type locality “Danzhou City” of the fungus.

In: *Penicillium* subgen. *Aspergilloides* sect. *Exilicaulis* ser. *Erubescentia*.

**Typification:** CHINA, Hainan Province, Danzhou City, Eman Town, 19°51′24″ N 109°13′54″ E, in tidal flat sediments, 2 September 2020, Hai-Yan Zhu, tt13610 (holotype HMAS 247880, ex-type strain CGMCC 3.25204).

**DNA barcodes:** ITS ON563150, *BenA* ON231295, *CaM* ON470838.

**Colony diameter:** 7 days, 25 °C (unless stated otherwise): CYA 21–24 mm; CYA 37 °C 6–7 mm; CYA 5 °C no growth; MEA 14–17 mm; PDA 17–19 mm; YES 12–15 mm.

**Colony characteristics:** On CYA 25 °C, 7 days: Colonies nearly circular, deep, wrinkled, protuberant at centers, radially sulcate; margins narrow, entire; mycelia white, light pink at center; texture velutinous to floccose; no sporulation; exudate colorless, no soluble pigments; reverse yellow. On CYA 37 °C, 7 days: Colonies nearly circular, plain, strongly wrinkled; margins narrow; mycelia white; texture velutinous to floccose; no sporulation; no exudate, no soluble pigments; reverse light yellow. On MEA 25 °C, 7 days: Colonies nearly circular, concave at centers, protuberant at margins; margins narrow to moderately wide, nearly entire; mycelia white, light grey at center; texture velutinous to floccose; sporulation sparse, light grey; no exudate, no soluble pigments; reverse yellow, white at periphery. On PDA 25 °C, 7 days: Colonies nearly circular, slightly protuberant at centers, edges irregular; margins narrow to moderately wide, nearly entire; mycelia white to cream, grey at center; texture velutinous to floccose; sporulation moderately dense, grey; no exudate, no soluble pigments; reverse light yellow to cream. On YES 25 °C, 7 days: Colonies nearly circular, deep, protuberant at centers; margins narrow, entire; mycelium white; texture velutinous to floccose; sporulation sparse; no exudate, no soluble pigments; reverse yellow, white at periphery.

**Micromorphology:** Conidiophores monoverticillate, rarely biverticillate; stipes smooth-walled, 12–40 × 2.0–3.0 µm; phialides flask-shaped to acerose, 2–5 per metula, 4.6–8.7 × 1.7–2.2 µm; conidia globose to subglobose, smooth to finely roughened, 2.2–3.0 µm in diam.

**Note:** This species is phylogenetically related to *P. catenatum* (Figure 3), but the latter differs in its larger phialides (8.0–12 × 2.5–3.0 µm) [67]. Moreover, the growth rates of *P. catenatum* were relatively slower than *P. danzhouense*, and the former inhabited the desert rather than tidal flat sediments.

***Penicillium tenue*** C. Liu, Z.Q. Zeng & W.Y. Zhuang, sp. nov. Figure 10.

**Fungal Names:** FN 571617.

**Etymology:** The specific epithet refers to the slender phialides.

In: *Penicillium* subgen. *Aspergilloides* sect. *Exilicaulis* ser. *Erubescentia*.

**Typification:** CHINA, Hainan Province, Danzhou City, Duntou Town, 19°09′06″ N 108°40′19″ E, in tidal flat sediments, 2 September 2020, Hai-Yan Zhu, tt13918 (holotype HMAS 247881, ex-type strain CGMCC 3.25205).

**DNA barcodes:** ITS ON563151, *BenA* ON231296, *CaM* ON470839, *RPB2* ON470842.

**Colony diameter:** 7 days, 25 °C (unless stated otherwise): CYA 11–12 mm; CYA 37 °C 12–13 mm; CYA 5 °C no growth; MEA 11–14 mm; PDA 12–13 mm; YES 11–12 mm.

**Colony characteristics:** On CYA 25 °C, 7 days: Colonies nearly circular, deep, protuberant at centers, radially sulcate; margins narrow, entire; mycelia white, light yellow at center; texture velutinous to floccose; no sporulation; exudate yellow, no soluble pigments; reverse yellow, deepens in the center. On CYA 37 °C, 7 days: Colonies nearly circular, concave at centers, protuberant at margins, slightly wrinkled, radially sulcate; margins narrow; mycelia white, light yellow at centers; texture velutinous to floccose; sporulation sparse; exudate yellow to yellowish-brown, no soluble pigments; reverse yellow, yellowish-brown in centers. On MEA 25 °C, 7 days: Colonies nearly circular, protuberant at centers; margins narrow to moderately wide, nearly entire; mycelia white, light yellow at center; texture velutinous to floccose; sporulation sparse; no exudate, no soluble pigments; reverse yellowish-brown, white at periphery. On PDA 25 °C, 7 days: Colonies nearly circular, protuberant at centers; margins narrow, nearly entire; mycelia white to yellow; texture velutinous to floccose; sporulation sparse; exudate yellow to yellowish-brown, no soluble pigments; reverse light yellow. On YES 25 °C, 7 days: Colonies nearly circular, deep, concave at centers, protuberant at margins; margins narrow, entire; mycelium white; texture velutinous to floccose; no sporulation; no exudate, no soluble pigments; reverse yellowish-brown, white at center and periphery.

**Micromorphology:** Conidiophores monoverticillate; stipes smooth-walled, 5.0–24 × 2.0–3.0 µm; phialides flask-shaped to acerose, 2–4 metula, 3.1–6.1 × 1.6–2.1 µm; conidia globose to subglobose, spinulose, 2.1–2.7 µm in diam.

**Note:** This species is phylogenetically related to *P. striatisporum* (Figure 3), but the latter differs in its ellipsoidal to ovoid and striate conidia [68]. Sequence comparisons between the ex-type cultures of the two species revealed that 8 bp, 12 bp, and 3 bp divergences were detected for the *BenA*, *CaM*, and *RPB2* regions.

***Penicillium zhanjiangense*** C. Liu, Z.Q. Zeng & W.Y. Zhuang, sp. nov. Figure 11.

**Fungal Names:** FN 571618.

**Etymology:** The specific epithet refers to the type locality “Zhanjiang City“ of the fungus.

In: *Penicillium* subgen. *Aspergilloides* section sect. *Lanata-Divaricata* ser. *Janthinella*.

**Typification:** CHINA, Guangdong Province, Zhanjiang City, Xuwen County, Southeast Village of China Mainland, 20°59′09″ N 109°40′53″ E, in tidal flat sediments, 30 August 2020, Hai-Yan Zhu, tt12003 (holotype HMAS 247882, ex-type strain CGMCC 3.25206).

**DNA barcodes:** ITS ON563149, *BenA* ON231294, *CaM* ON470837.

**Colony diameter:** 7 days, 25 °C (unless stated otherwise): CYA 28–30 mm; CYA 37 °C 42–43 mm; CYA 5 °C no growth; MEA 41–48 mm; PDA 36–37 mm; YES 22–24 mm.

**Colony characteristics:** On CYA 25 °C, 7 days: Colonies nearly circular, concave at centers, protuberant at margins, wrinkled, radially sulcate; margins narrow to moderately wide, nearly entire; mycelia white, light grey at center; texture velutinous to floccose; sporulation sparse; conidia en masse cream to light grey; no exudate, no soluble pigments; reverse yellowish-brown, light green to blackish-green at centers, white at periphery. On CYA 37 °C, 7 days: Colonies nearly circular, concave at centers, protuberant at margins, wrinkled, radially sulcate; margins narrow; mycelia pink, white to yellowish-grey at centers; texture velutinous; texture velutinous to floccose; sporulation sparse; exudate pink to pinkish-brown, no soluble pigments; reverse pink to dark pink, yellow at periphery. On MEA 25 °C, 7 days: Colonies nearly circular, concave at centers, protuberant at margins, wrinkled, radially sulcate; margins narrow to moderately wide, nearly entire; mycelia cream, brown at center; texture velutinous to floccose; sporulation dense; conidia en masse brownish-yellow; no exudate, no soluble pigments; reverse yellowish-brown, deepens in the center. On PDA 25 °C, 7 days: Colonies nearly circular, concave at centers, protuberant at margins, slightly wrinkled, radially sulcate; margins narrow to moderately wide, nearly entire; mycelia light yellow, brown at center; texture velutinous to floccose; sporulation dense; conidia en masse brownish-yellow; no exudate, no soluble pigments; reverse light green to dark green, deepens in the center. On YES 25 °C, 7 days: Colonies nearly circular, deep, concave at centers, protuberant at margins, slightly wrinkled, radially sulcate; margins narrow, nearly entire; mycelium white to cream; texture velutinous to floccose; sporulation sparse; no exudate, no soluble pigments; reverse yellow, white at periphery.

**Micromorphology:** Conidiophores monoverticillate to biverticillate; stipes smooth-walled, 24–170 × 2.4–3.7 µm; metulae cylindrical, 8.1–28.3 × 2.2–3.1 µm; phialides flask-shaped to acerose, tapering into thin neck, 2–3 per metula, 4.9–15.5 × 1.9–2.9 µm; conidia globose to subglobose, smooth to finely roughened, 2.0–3.3 µm in diam.

**Note:** This species is phylogenetically related to *P. janthinellum* (Figure 4), but the latter differs in its faster growth rate on YES (44–46 mm) at 25 °C, while a slower growth rate was observed on CYA at 37 °C (20–30 mm) [69].

***Talaromyces virens*** C. Liu, Z.Q. Zeng & W.Y. Zhuang, sp. nov. Figure 12.

**Fungal Names:** FN 571619.

**Etymology:** The specific epithet refers to the green conidia.

In: *Talaromyces* section *Talaromyces*.

**Typification:** CHINA, Hainan Province, Wenchang City, Dongjiao Town, 110°50′40″ N 19°32′27″ E, in tidal flat sediments, 1 September 2020, Hai-Yan Zhu, tt13401 (holotype HMAS 247883, ex-type strain CGMCC 3.25207).

**DNA barcodes:** ITS ON563152, *BenA* ON231297, *CaM* ON470840, *RPB2* ON470841.

**Colony diameter:** 7 days, 25 °C (unless stated otherwise): CYA 19–20 mm; CYA 37 °C 6–7 mm; CYA 5 °C no growth; MEA 23–26 mm; PDA 22–23 mm; YES 18–19 mm.

**Colony characteristics:** On CYA 25 °C, 7 days: Colonies nearly circular, protuberant in centers; margins narrow to moderately wide, nearly entire; mycelia white; texture velutinous; sporulation dense; conidia en masse dark olive green; no exudate, no soluble pigments; reverse light khaki, light brown at centers, light yellow and white at periphery. On CYA 37 °C, 7 days: Colonies nearly circular, deep, wrinkled, deeply concave in centers; margins narrow; mycelia white; texture velutinous; sporulation dense; conidia en masse grey; no exudate, no soluble pigments; reverse grey or white. On MEA 25 °C, 7 days: Colonies nearly circular, slightly protuberant in centers; margins wide, nearly entire; mycelia white; texture velutinous; sporulation dense in center, olive-drab; no exudate, no soluble pigments; reverse light yellow, light orange at centers. On PDA 25 °C, 7 days: Colonies nearly circular, protuberant in centers; margins wide, nearly entire; mycelia white; texture velutinous to floccose; sporulation dense in center, deep green; no exudate, no soluble pigments; reverse white, light coral to red at centers, white at periphery. On YES 25 °C, 7 days: Colonies nearly circular, protuberant in centers; margins narrow to moderately wide, nearly entire; mycelium gray-green to yellowish-green at center, white at margin; texture velutinous to floccose; sporulation moderately dense, yellow to green; no exudate, no soluble pigments; reverse white, dark yellow at centers.

**Micromorphology:** Conidiophores biverticillate; stipes smooth-walled, 147–220 × 2.4–4.0 µm; metulae cylindrical, 9.1–14.1 × 2.5–3.4 µm; phialides flask-shaped to acerose, tapering into thin neck, 2–6 per metula, 7.7–11.2 × 2.2–3.3 µm; conidia globose to subglobose, green, 2.4–4.0 µm in diam.

**Note:** This species is morphologically and phylogenetically related to *T. xishaensis* (Figure 5). However, the latter has greyish-green colonies on CYA, yellowish-green colonies on MEA, and grey to bluish-green colonies on YES [70].

## 4. Discussion

The genus *Aspergillus* is divided into six subgenera with 27 sections [13,22]. Our new species *A. liaoningensis* was well-located in the ser. *Candidi* of sect. *Candidi* (BIPP/MLBS = 100%/100%) (Figure 1). Many species within this section have been reported as producers of secondary metabolites, such as taichunamides, shikimic acid derivatives, and terpene-derived taichunins [71,72,73]. Studies on the metabolic application of *A. liaoningensis* are surely our future goal. *Aspergillus plumeriae* belongs to ser. *Spelaei* of sect. *Flavipedes*, which is in accordance with the growth rate at 37 °C [65]. *Aspergillus subinflatus* was classified as a member of ser. *Inflati*, sect. *Cremei* of subgen. *Cremei*, and is most related to *A. inflatus* in both phylogeny and morphology. However, sequence comparisons revealed that there were 69 bp, 74 bp, and 88 bp unmatched loci detected in the *BenA*, *CaM,* and *RPB2* regions between them.

Both *P. danzhouense* and *P. tenue* form monoverticillate conidiophores, consistent with the other members of ser. *Erubescentia* in sect. *Exilicaulis* [7,48,74,75]. The phylogenetic results indicate that *P. danzhouense* is closely related to *P. catenatum* with high statistical support values (Figure 3); however, the latter differs in having larger phialides (8.0–12 × 2.5–3.0 µm vs. 4.6–8.7 × 1.7–2.2 µm) [67]. Moreover, there were 20 bp and 56 bp divergences in the *BenA* and *CaM* regions between them. Our results also showed that *P. tenue* was grouped with *P. striatisporum*, receiving full support (Figure 3), but the latter possesses ellipsoidal to ovoid and striate conidia [68]. *Penicillium dravuni*, a marine-derived species belonging to this section, was not included in our phylogenetic analysis because no sequence data are available at present. However, it can be easily distinguished from *P. danzhouense* (white to light pink colonies) and *P. tenue* (white to light yellow colonies) because it forms yellow-gray colonies and has faster growth rates (25–35 mm vs. 21–24 mm, and 11–12 mm in CYA at 25 °C) [76].

Sect. *Lanata-Divaricata* is a species-rich section in *Penicillium*, and about 85 species have been described hitherto [4,20,27,28,31,33]. Our phylogenetic tree indicated that *P. zhanjiangense* was well-located among other species of sect. *Lanata-Divaricata*, with high supporting values (BIPP/MLBS = 97%/100%), and it clustered with *P. janthinellum* (BIPP/MLBS = 100%/100%) (Figure 4). A new species of the section was added in this study.

Since the establishment of *Talaromyces*, eight sections have been proposed: *Bacillispori*, *Helici*, *Islandici*, *Purpurei*, *Subinflati*, *Talaromyces*, *Tenues*, and *Trachyspermi* [40]. The phylogenetic overview of the sect. *Talaromyces* was conducted by Wang and Zhuang [40], and about 88 species are currently known in this section [40,41,42]. The three-locus phylogeny formed a well-supported monophyletic group (BIPP/MLBS = 100%/100%) and indicates that *T. virens* is related to but distinct from *T. xishaensis* (Figure 5). They can be easily distinguished by their different colony features on different media [70]. Moreover, there are 11 bp, 34 bp, and 32 bp unmatched loci detected in the *BenA*, *CaM*, and *RPB2* regions between the ex-type cultures (CGMCC 3.25207 and CGMCC 3.17995).

Species of the genera *Penicillium*, *Aspergillus*, and *Talaromyces* have been isolated from various substrates, including dust, soil, dung, cloth, human tissue, plants, and insects [4,11]. Due to the special ecological habitat, the fungi of these groups within tidal flats have high biodiversity and are an important source of active natural products [77,78,79,80]. Recently, three novel taxa of *Penicillium* were reported in tidal flats [28]. Similarly, the present study introduces three species of *Aspergillus* as well as three taxa of *Penicillium* and one of *Talaromyces* derived from tidal flat sediments. With the extensive use of molecular approaches, large-scale surveys in these unexplored tidal flats regions will significantly improve our knowledge of fungal species diversity in special ecological environments.

## 5. Conclusions

The filamentous fungi from tidal flat sediments in China were surveyed, and seven novel taxa of the genera *Aspergillus*, *Penicillium*, and *Talaromyces* were discovered. With the joining of the new species, the phylogenetic relationships among species of these three genera were updated.

## Figures and Tables

**Figure 1 jof-09-00960-f001:**
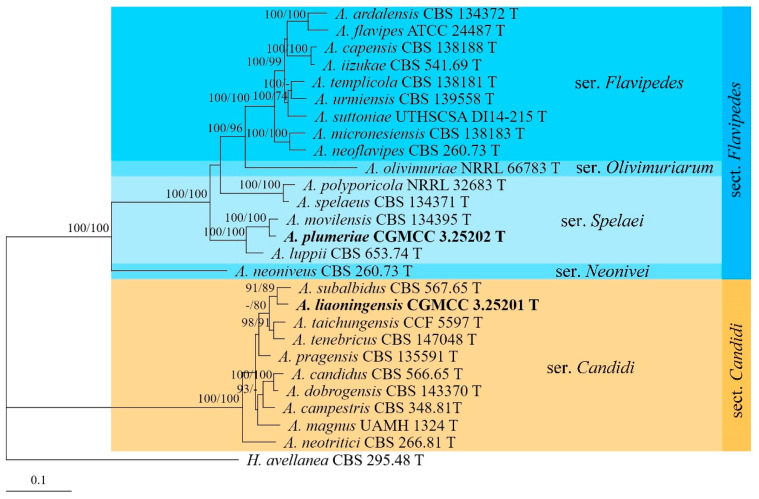
BI tree generated from analyses of combined *BenA*, *CaM*, and *RPB2* sequences of *Aspergillus* sect. *Candidi* and *Flavipedes*. BIPP ≥ 90% (left) and MLBS ≥ 70% (right) are indicated at nodes.

**Figure 2 jof-09-00960-f002:**
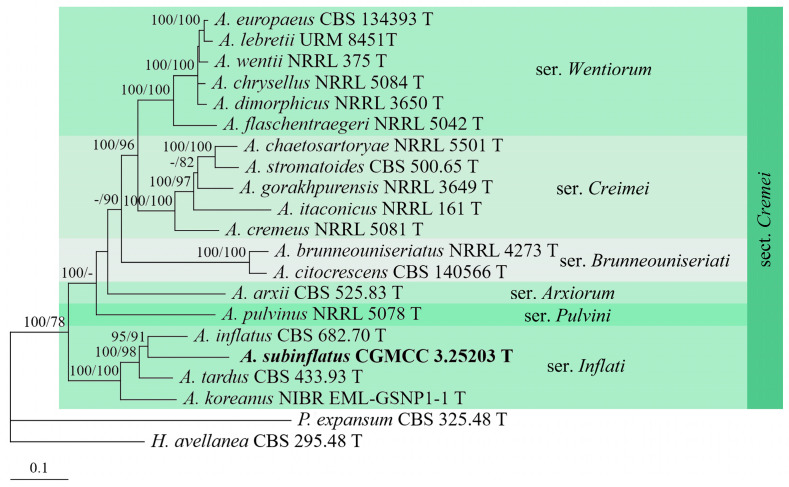
BI tree generated from analyses of combined *BenA*, *CaM*, and *RPB2* sequences of *Aspergillus* sect. *Cremei* species. BIPP ≥ 90% (left) and MLBS ≥ 70% (right) are indicated at nodes.

**Figure 3 jof-09-00960-f003:**
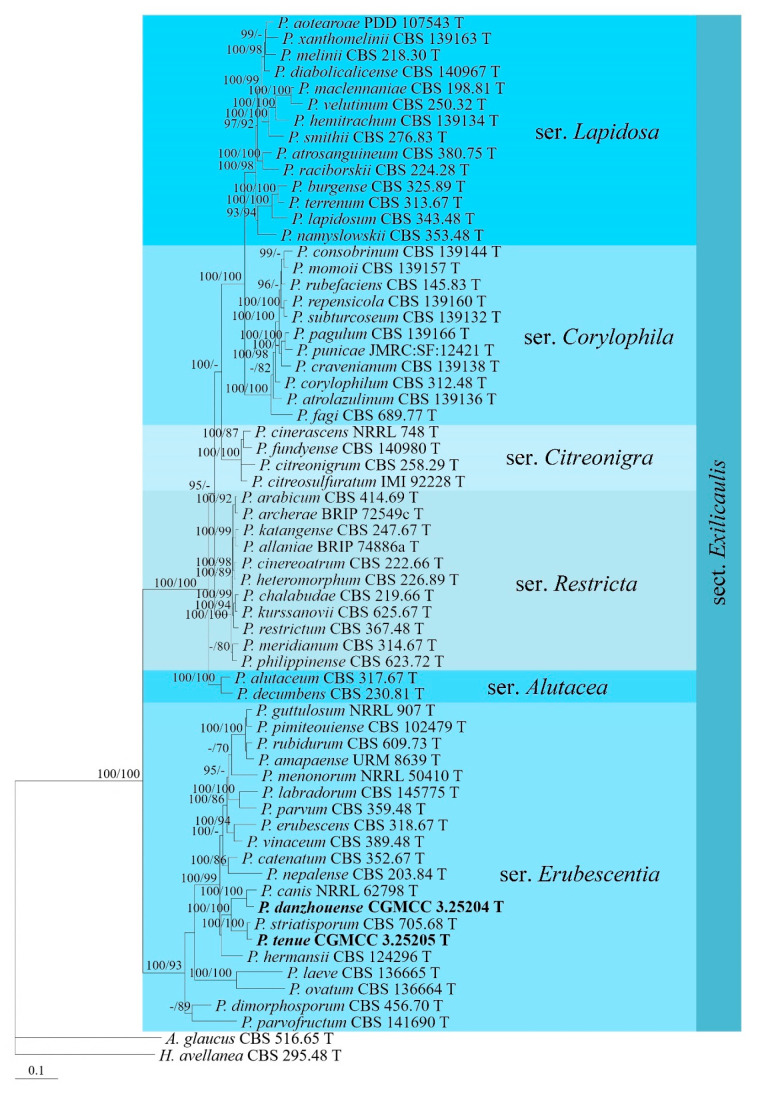
BI tree generated from analyses of combined *BenA*, *CaM*, and *RPB2* sequences of *Penicillium* sect. *Exilicaulis* species. BIPP ≥ 90% (left) and MLBS ≥ 70% (right) are indicated at nodes.

**Figure 4 jof-09-00960-f004:**
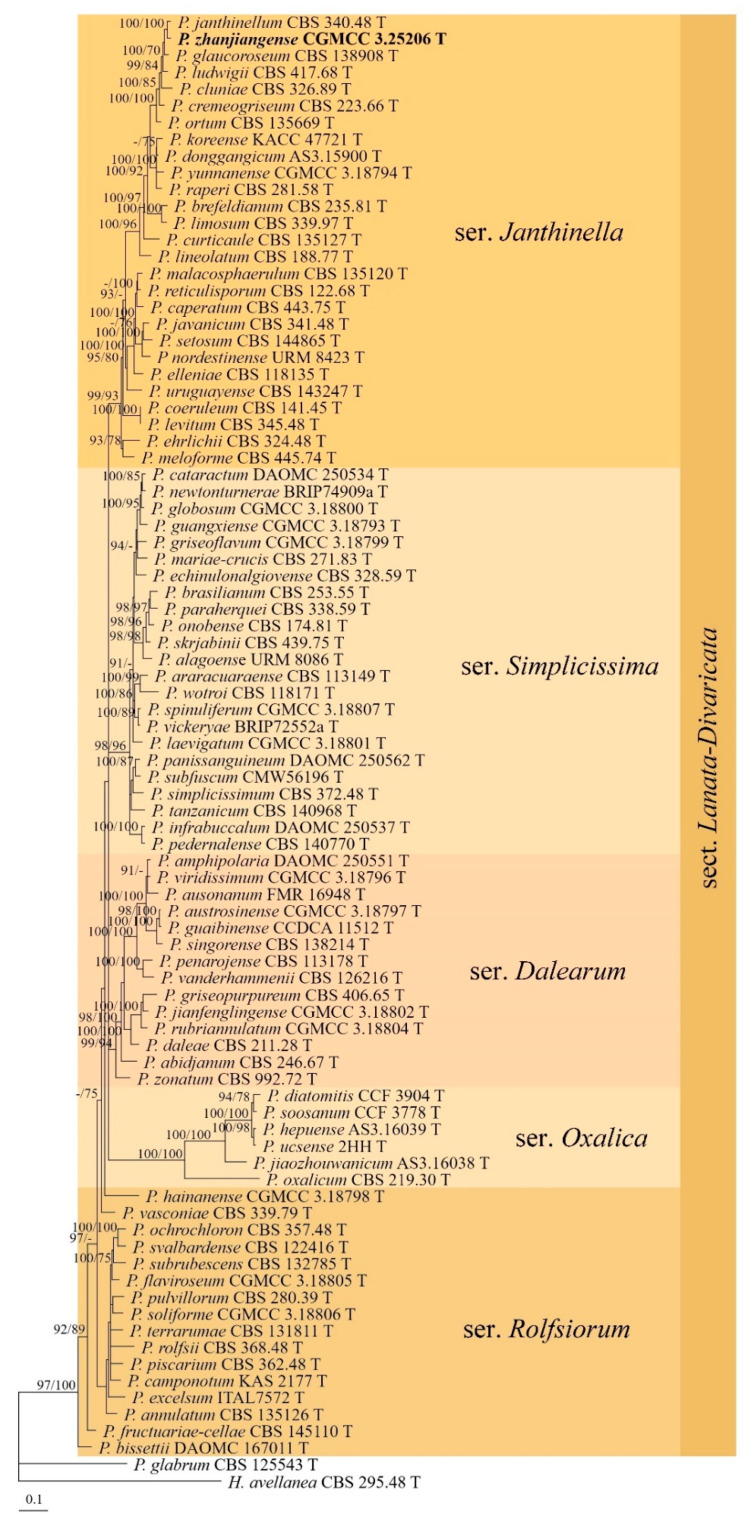
BI tree generated from analyses of combined *BenA*, *CaM*, and *RPB2* sequences of *Penicillium* sect. *Lanata-Divaricata* species. BIPP ≥ 90% (left) and MLBS ≥ 70% (right) are indicated at nodes.

**Figure 5 jof-09-00960-f005:**
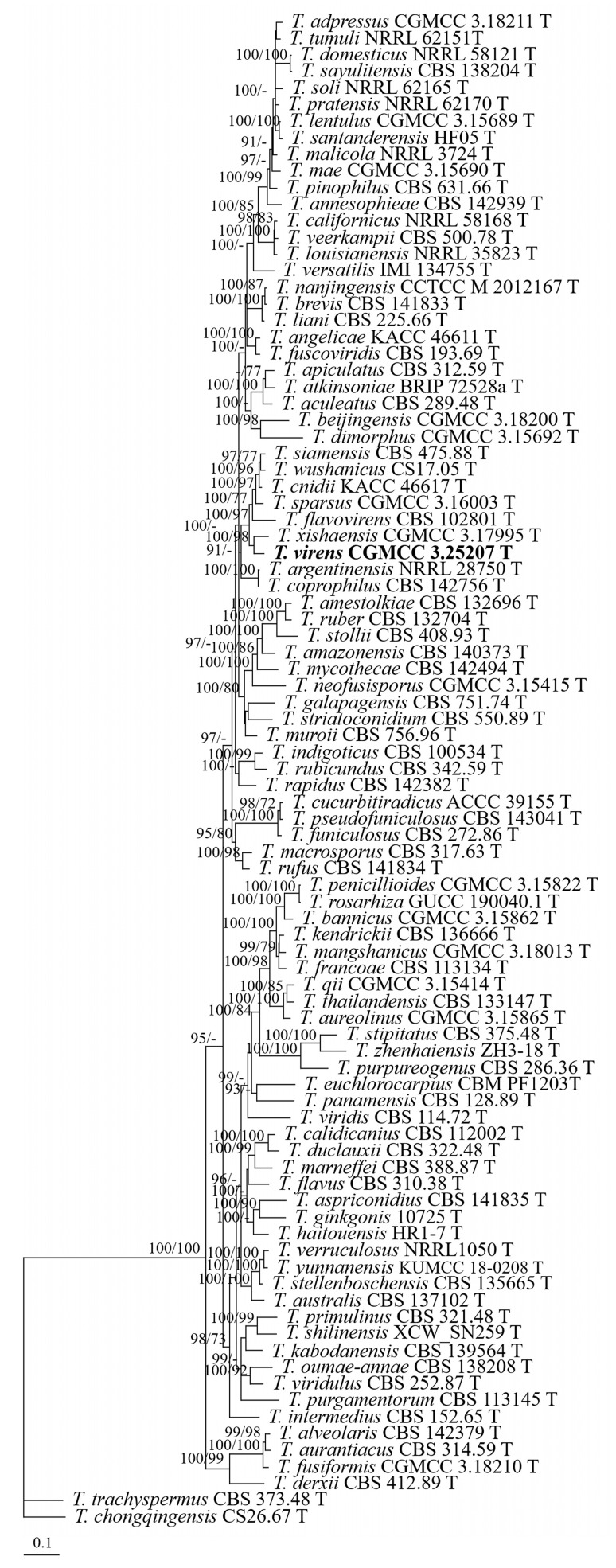
BI tree generated from analyses of combined *BenA*, *CaM*, and *RPB2* sequences of *Talaromyces* sect. *Talaromyces* species. BIPP ≥ 90% (left) and MLBS ≥ 70% (right) are indicated at nodes.

**Figure 6 jof-09-00960-f006:**
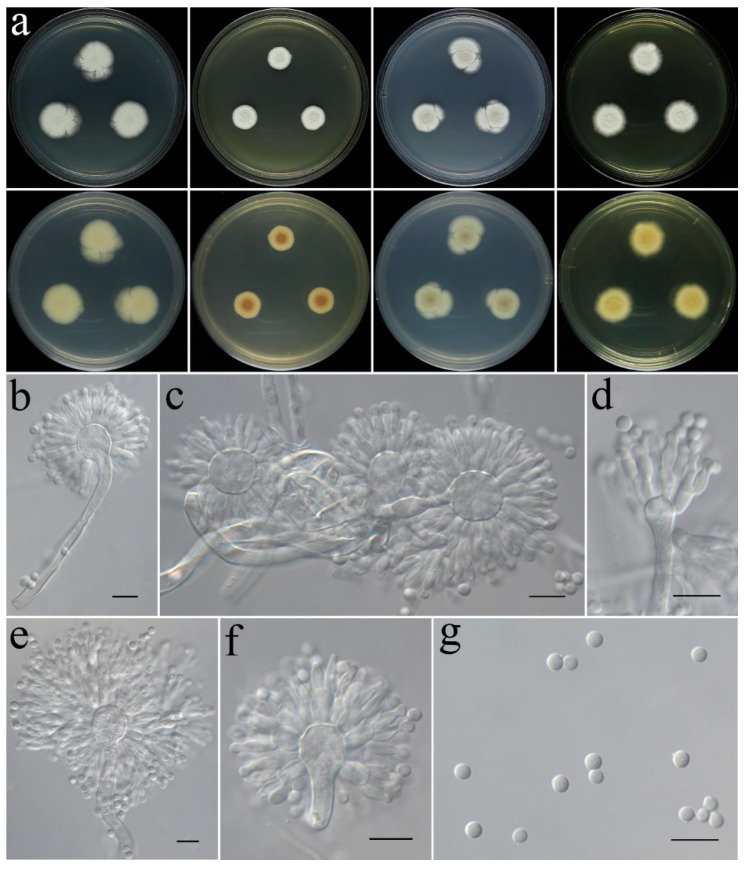
Colonial and microscopic morphology of *Aspergillus liaoningensis* (CGMCC 3.25201). (**a**) Colonies after 7 days at 25 °C; top row left to right: obverse CYA, MEA, PDA, and YES; bottom row left to right: reverse CYA, MEA, PDA, and YES; (**b**–**f**) Conidiophores; (**g**) Conidia. Scale bars: (**b**–**g**) = 10 μm.

**Figure 7 jof-09-00960-f007:**
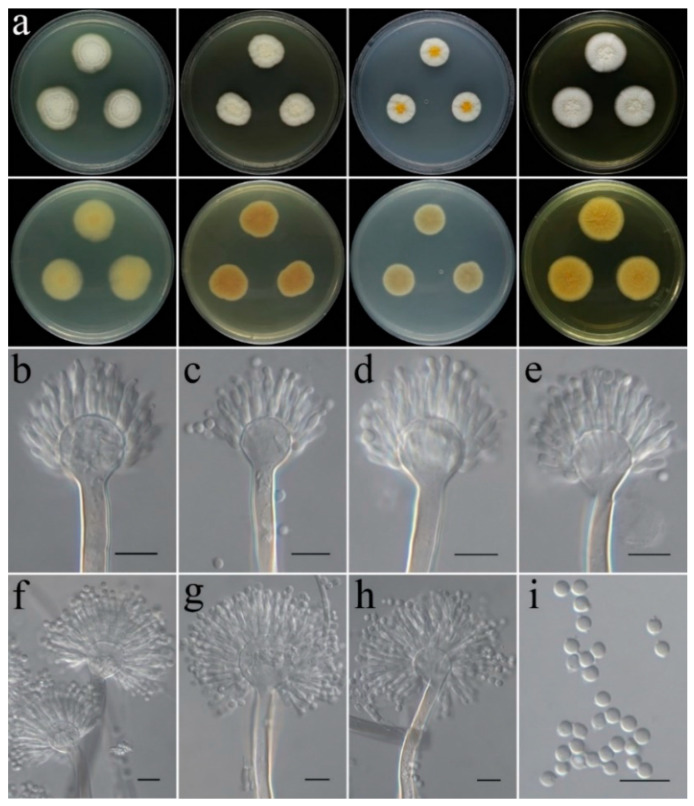
Colonial and microscopic morphology of *Aspergillus plumeriae* (CGMCC 3.25202). (**a**) Colonies after 7 days at 25 °C; top row left to right: obverse CYA, MEA, PDA, and YES; bottom row left to right: reverse CYA, MEA, PDA, and YES; (**b**–**h**) Conidiophores; (**i**) Conidia. Scale bars: (**b**–**i**) = 10 μm.

**Figure 8 jof-09-00960-f008:**
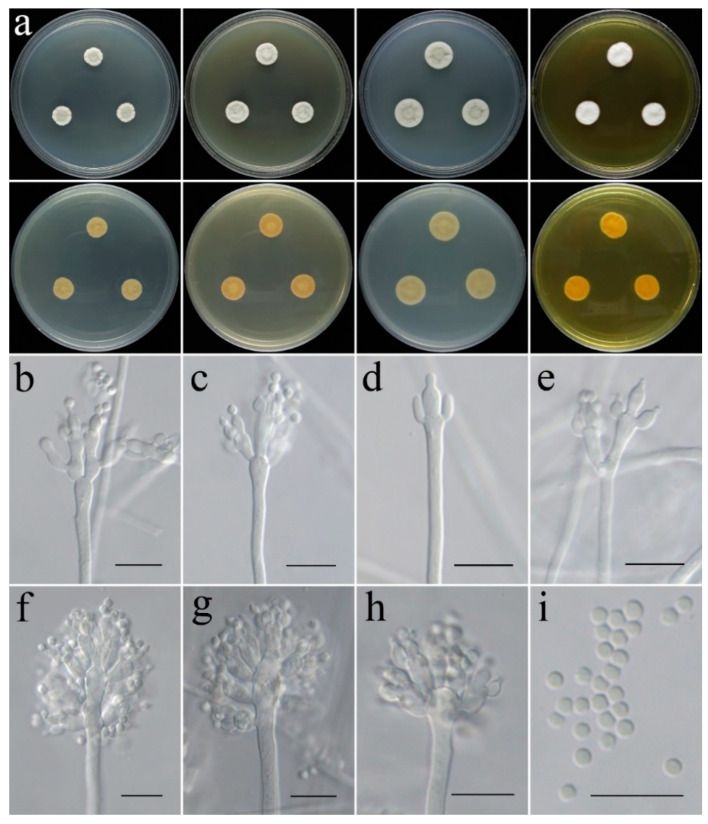
Colonial and microscopic morphology of *Aspergillus subinflatus* (CGMCC 3.25203). (**a**) Colonies after 7 days at 25 °C; top row left to right: obverse CYA, MEA, PDA, and YES; bottom row left to right: reverse CYA, MEA, PDA, and YES; (**b**–**h**) Conidiophores; (**i**) Conidia. Scale bars: (**b**–**i**) = 10 μm.

**Figure 9 jof-09-00960-f009:**
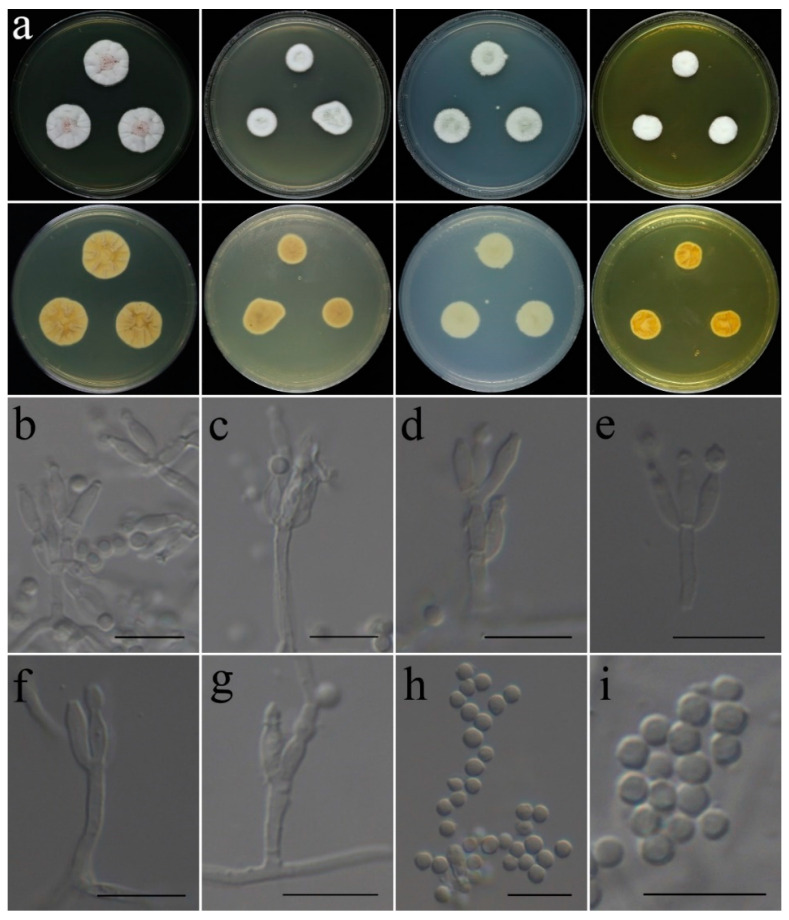
Colonial and microscopic morphology of *Penicillium danzhouense* (CGMCC 3.25204). (**a**) Colonies after 7 days at 25 °C; top row left to right: obverse CYA, MEA, PDA, and YES; bottom row left to right: reverse CYA, MEA, PDA, and YES; (**b**–**g**) Conidiophores; (**h**,**i**) Conidia. Scale bars: (**b**–**i**) = 10 μm.

**Figure 10 jof-09-00960-f010:**
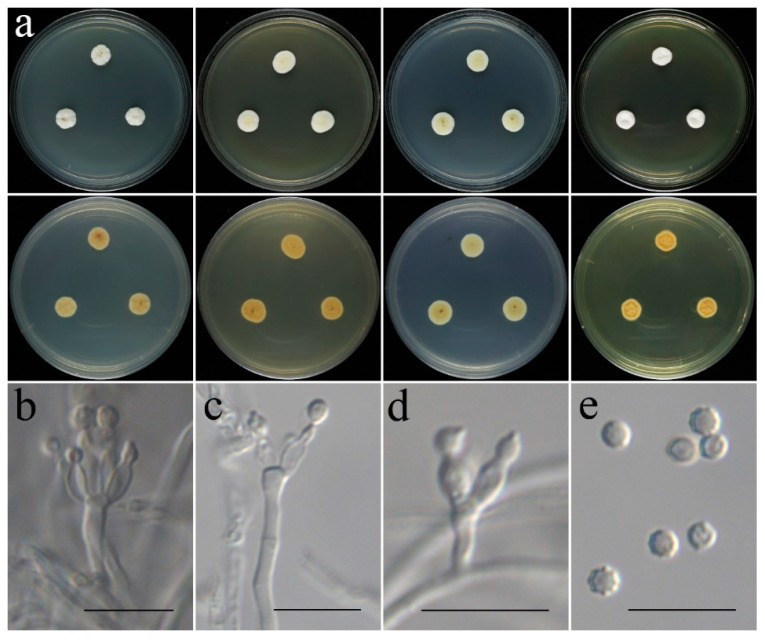
Colonial and microscopic morphology of *Penicillium tenue* (CGMCC 3.25205). (**a**) Colonies after 7 days at 25 °C; top row left to right: obverse CYA, MEA, PDA, and YES; bottom row left to right: reverse CYA, MEA, PDA, and YES; (**b**–**d**) Conidiophores; (**e**) Conidia. Scale bars: (**b**–**e**) = 10 μm.

**Figure 11 jof-09-00960-f011:**
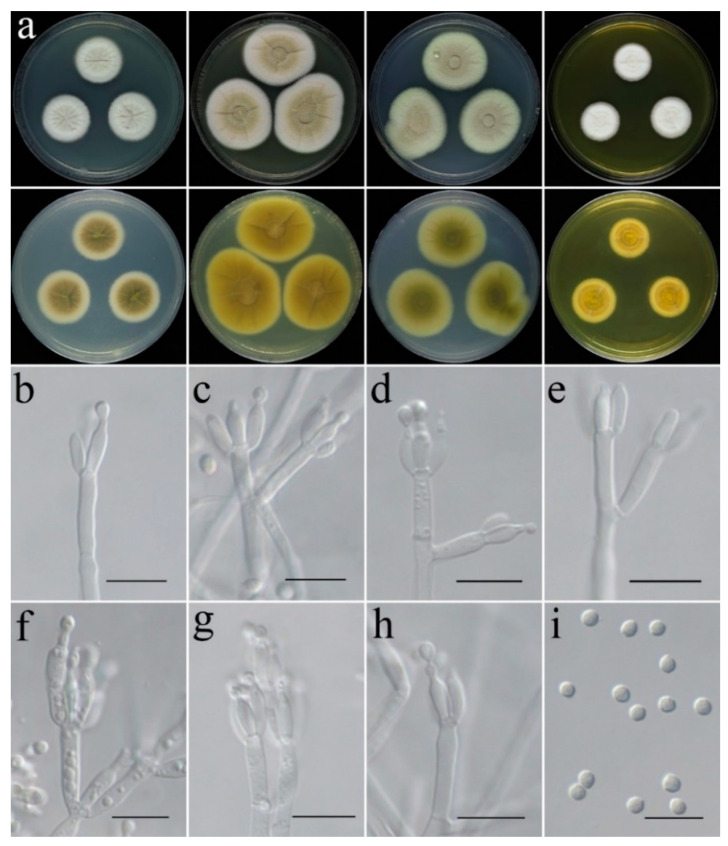
Colonial and microscopic morphology of *Penicillium zhanjiangense* (CGMCC 3.25206). (**a**) Colonies after 7 days at 25 °C; top row left to right: obverse CYA, MEA, PDA, and YES; bottom row left to right: reverse CYA, MEA, PDA, and YES; (**b**–**h**) Conidiophores; (**i**) Conidia. Scale bars: (**b**–**i**) = 10 μm.

**Figure 12 jof-09-00960-f012:**
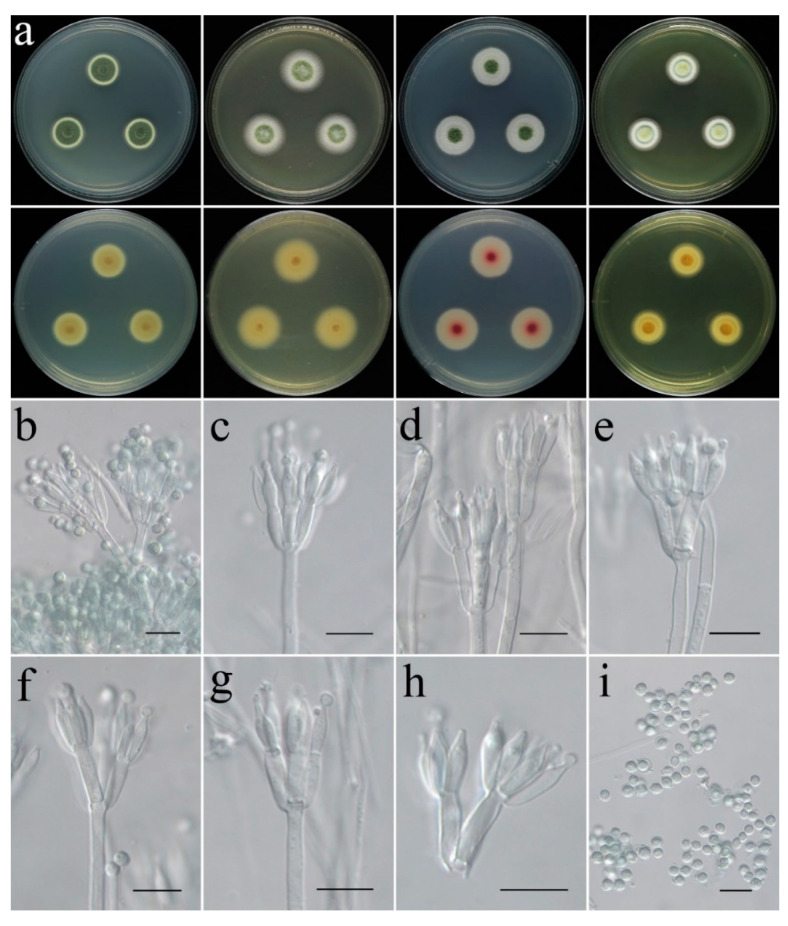
Colonial and microscopic morphology of *Talaromyces virens* (CGMCC 3.25207). (**a**) Colonies after 7 days at 25 °C; top row left to right: obverse CYA, MEA, PDA, and YES; bottom row left to right: reverse CYA, MEA, PDA, and YES; (**b**–**h**) Conidiophores; (**i**) Conidia. Scale bars: (**b**–**i**) = 10 μm.

**Table 1 jof-09-00960-t001:** Names, strain numbers, and corresponding GenBank accession numbers of the taxa used in this study.

Species	Strain Numbers	ITS	*BenA*	*CaM*	*RPB2*
sect. *Candidi*					
*A. campestris*	CBS 348.81 T	EF669577	EU014091	EF669535	EF669619
*A. candidus*	CBS 566.65 T	EF669592	EU014089	EF669550	EF669634
*A. dobrogensis*	CBS 143370 T	LT626959	LT627027	LT558722	LT627028
*A. liaoningensis*	CGMCC 3.25201 T	**ON563148**	**ON231293**	**ON470836**	**ON470844**
*A. magnus*	UAMH 1324 T	ON156376	ON164570	ON164619	ON164517
*A. neotritici*	CCF 3853 T	FR727136	FR775327	HE661598	LT627021
*A. pragensis*	CBS 135591 T	FR727138	HE661604	FR751452	LN849445
*A. subalbidus*	CBS 567.65 T	EF669593	KP987050	EF669551	EF669635
*A. taichungensis*	CCF 5597 T	LT626957	EU076297	HG916679	LT627016
*A. tenebricus*	CBS 147048 T	ON156389	ON164584	ON164623	ON164532
sect. *Flavipedes*					
*A. ardalensis*	CBS 134372 T	FR733808	HG916683	HG916725	HG916704
*A. capensis*	CBS 138188 T	KJ775550	KJ775072	KJ775279	KP987020
*A. flavipes*	ATCC 24487 T	EF669591	EU014085	EF669549	EF669633
*A. iizukae*	CBS 541.69 T	EF669597	EU014086	EF669555	EF669639
*A. luppii*	CBS 653.74 T	EF669617	EU014079	EF669575	EF669659
*A. micronesiensis*	CBS 138183 T	KJ775548	KJ775085	KP987067	KP987023
*A. movilensis*	CBS 134395 T	KP987089	HG916697	HG916740	HG916718
*A. neoflavipes*	CBS 260.73 T	EF669614	EU014084	EF669572	EF669656
*A. neoniveus*	CBS 261.73 T	EF669612	EU014098	EF669570	KP987024
*A. olivimuriae*	NRRL 66783	MH298877	MH492010	MH492011	MH492012
*A. plumeriae*	CGMCC 3.25202 T	**ON563147**	**ON231292**	**ON470835**	**ON470843**
*A. polyporicola*	NRRL 32683 T	EF669595	EU014088	EF669553	EF669637
*A. spelaeus*	CBS 134371 T	HG915905	HG916698	HG916741	HG916719
*A. suttoniae*	UTHSCSA DI14-215	LT899487	LT899536	LT899589	LT899644
*A. templicola*	CBS 138181 T	KJ775545	KJ775092	KJ775394	KP987017
*A. urmiensis*	CBS 139558 T	KP987073	KP987041	KP987056	KP987030
sect. *Cremei*					
*A. arxii*	CBS 525.83 T	MN431361	MN969365	MN969223	JN121529
*A. brunneouniseriatus*	NRRL 4273 T	EF652141	EF652123	EF652138	EF652089
*A. chaetosartoryae*	NRRL 5501 T	EF652144	EF652117	EF652129	EF652099
*A. chrysellus*	NRRL 5084 T	EF652155	EF652109	EF652136	EF652090
*A. citocrescens*	CBS 140566 T	FR727121	FR775317	LN878969	MN969163
*A. cremeus*	NRRL 5081 T	EF652149	EF652120	EF652125	EF652101
*A. dimorphicus*	NRRL 3650 T	EF652154	EF652111	EF652135	EF652096
*A. europaeus*	CBS 134393T	LN908996	LN909006	LN909007	LT548274
*A. flaschentraegeri*	NRRL 5042 T	EF652150	EF652113	EF652130	EF652102
*A. gorakhpurensis*	NRRL 3649 T	EF652145	EF652114	EF652126	EF652097
*A. inflatus*	CBS 682.70 T	FJ531054	FJ531008	FJ531090	JN406529
*A. itaconicus*	NRRL 161 T	EF652147	EF652118	EF652140	EF652103
*A. koreanus*	NIBR EML-GSNP1-1	KX216525	KX216530	KX216528	KX216531
*A. lebretii*	URM 8451 T	ON862928	OP672381	OP290539	OP290510
*A. pulvinus*	NRRL 5078 T	EF652159	EF652121	EF652139	EF652104
*A. stromatoides*	CBS 500.65 T	EF652146	FJ531038	EF652127	EF652098
*A. subinflatus*	CGMCC 3.25203 T	**ON563146**	**ON231291**	**ON470834**	**ON470845**
*A. tardus*	CBS 433.93 T	FJ531045	FJ531001	FJ531084	n.a.
*A. wentii*	NRRL 375 T	EF652151	EF652106	EF652131	EF652092
Sect. *Lanata-divaricata*					
*P. abidjanum*	CBS 246.67 T	GU981582	GU981650	KF296383	JN121469
*P. alagoense*	URM 8086 T	MK804503	MK802333	MK802336	MK802338
*P. amphipolaria*	DAOMC 250551 T	KT887872	KT887833	KT887794	n.a.
*P. annulatum*	CBS 135126 T	JX091426	JX091514	JX141545	KF296410
*P. araracuaraense*	CBS 113149 T	GU981597	GU981642	KF296373	KF296414
*P. ausonanum*	FMR 16948 T	LR655808	LR655809	LR655810	LR655811
*P. austrosinense*	CGMCC 3.18797 T	KY495007	KY495116	KY494947	KY495061
*P. bissettii*	DAOMC 167011 T	KT887845	KT887806	KT887767	MN969107
*P. brasilianum*	CBS 253.55 T	GU981577	GU981629	MN969239	KF296420
*P. brefeldianum*	CBS 235.81 T	AF033435	GU981623	EU021683	KF296421
*P. camponotum*	KAS 2177 T	KT887855	KT887816	KT887777	MN969179
*P. caperatum*	CBS 443.75 T	KC411761	GU981660	KF296392	KF296422
*P. cataractum*	DAOMC 250534 T	KT887847	KT887808	KT887769	n.a.
*P. cluniae*	CBS 326.89 T	KF296406	KF296471	KF296402	KF296424
*P. coeruleum*	CBS 141.45 T	GU981606	GU981655	KF296393	KF296425
*P. cremeogriseum*	CBS 223.66 T	GU981586	GU981624	KF296403	KF296426
*P. curticaule*	CBS 135127 T	FJ231021	JX091526	JX141536	KF296417
*P. daleae*	CBS 211.28 T	GU981583	GU981649	KF296385	KF296427
*P. diatomitis*	CCF 3904 T	FJ430748	HE651133	LT970912	LT797560
*P. donggangicum*	AS3.15900 T	MW946996	MZ004914	MZ004918	MW979253
*P. echinulonalgiovense*	CBS 328.59 T	GU981587	GU981631	KX961269	KX961301
*P. ehrlichii*	CBS 324.48 T	AF033432	GU981652	KF296395	KF296428
*P. elleniae*	CBS 118135 T	GU981612	GU981663	KF296389	KF296429
*P. excelsum*	ITAL7572 T	KR815341	KP691061	KR815342	MN969166
*P. flaviroseum*	CGMCC 3.18805 T	KY495032	KY495141	KY494972	KY495083
*P. fructuariae-cellae*	CBS 145110 T	MK039434	KU554679	MK045337	n.a.
*P. glaucoroseum*	CBS 138908 T	MN431390	MN969383	MN969257	MN969119
*P. globosum*	CGMCC 3.18800 T	KY495014	KY495123	KY494954	KY495067
*P. griseoflavum*	CGMCC 3.18799 T	KY495011	KY495120	KY494951	KY495064
*P. griseopurpureum*	CBS 406.65 T	KF296408	KF296467	KF296384	KF296431
*P. guaibinense*	CCDCA 11512 T	MH674389	MH674391	MH674393	n.a.
*P. guangxiense*	CGMCC 3.18793 T	KY494986	KY495095	KY494926	n.a.
*P. hainanense*	CGMCC 3.18798 T	KY495009	KY495118	KY494949	n.a.
*P. hepuense*	AS3.16039 T	MW946994	MZ004912	MZ004916	MW979254
*P. infrabuccalum*	DAOMC 250537 T	KT887856	KT887817	KT887778	n.a.
*P. janthinellum*	CBS 340.48 T	GU981585	GU981625	KF296401	JN121497
*P. javanicum*	CBS 341.48 T	GU981613	GU981657	KF296387	JN121498
*P. jianfenglingense*	CGMCC 3.18802 T	KY495016	KY495125	KY494956	KY495069
*P. jiaozhouwanicum*	AS3.16038 T	MW946993	MZ004911	MZ004915	MW979252
*P. koreense*	KACC 47721 T	KJ801939	KM000846	MN969317	MN969159
*P. laevigatum*	CGMCC 3.18801 T	KY495015	KY495124	KY494955	KY495068
*P. levitum*	CBS 345.48 T	GU981607	GU981654	KF296394	KF296432
*P. limosum*	CBS 339.97 T	GU981568	GU981621	KF296398	KF296433
*P. lineolatum*	CBS 188.77 T	GU981579	GU981620	KF296397	KF296434
*P. ludwigii*	CBS 417.68 T	KF296409	KF296468	KF296404	KF296435
*P. malacosphaerulum*	CBS 135120 T	FJ231026	JX091524	JX141542	KF296438
*P. mariae-crucis*	CBS 271.83 T	GU981593	GU981630	KF296374	KF296439
*P. meloforme*	CBS 445.74 T	KC411762	GU981656	KF296396	KF296440
*P. newtonturnerae*	BRIP74909a T	OP903478	OP921964	OP921962	OP921963
*P. nordestinense*	URM 8423 T	OV265270	OV265324	OV265272	OM927721
*P. ochrochloron*	CBS 357.48 T	GU981604	GU981672	KF296378	KF296445
*P. onobense*	CBS 174.81 T	GU981575	GU981627	KF296371	KF296447
*P. ortum*	CBS 135669 T	JX091427	JX091520	JX141551	KF296443
*P. oxalicum*	CBS 219.30 T	AF033438	KF296462	KF296367	JN121456
*P. panissanguineum*	DAOMC 250562 T	KT887862	KT887823	KT887784	n.a.
*P. paraherquei*	CBS 338.59 T	AF178511	KF296465	KF296372	KF296449
*P. pedernalense*	CBS 140770 T	KU255398	KU255396	MN969322	MN969184
*P. penarojense*	CBS 113178 T	GU981570	GU981646	KF296381	KF296450
*P. piscarium*	CBS 362.48 T	GU981600	GU981668	KF296379	KF296451
*P. pulvillorum*	CBS 280.39 T	AF178517	GU981670	KF296377	KF296452
*P. raperi*	CBS 281.58 T	AF033433	GU981622	KF296399	KF296453
*P. reticulisporum*	CBS 122.68 T	AF033437	GU981665	KF296391	KF296454
*P. rolfsii*	CBS 368.48 T	JN617705	GU981667	KF296375	KF296455
*P. rubriannulatum*	CGMCC 3.18804 T	KY495029	KY495138	KY494969	KY495080
*P. setosum*	CBS 144865 T	KT852579	MF184995	MH105905	MH016196
*P. simplicissimum*	CBS 372.48 T	GU981588	GU981632	KF296368	JN121507
*P. singorense*	CBS 138214 T	KJ775674	KJ775167	KJ775403	n.a.
*P. skrjabinii*	CBS 439.75 T	GU981576	GU981626	KF296370	EU427252
*P. soliforme*	CGMCC 3.18806 T	KY495038	KY495147	KY494978	n.a.
*P. soosanum*	CCF 3778 T	FJ430745	FM865811	LT970913	LT797561
*P. spinuliferum*	CGMCC 3.18807 T	KY495040	KY495149	KY494980	KY495090
*P. subfuscum*	CMW56196 T	MT949907	MT957412	MT957454	MT957480
*P. subrubescens*	CBS 132785 T	KC346350	KC346327	KC346330	KC346306
*P. svalbardense*	CBS 122416 T	GU981603	KC346325	KC346338	KF296457
*P. tanzanicum*	CBS 140968 T	KT887841	KT887802	KT887763	MN969183
*P. terrarumae*	CBS 131811 T	MN431397	KX650295	MN969323	MN969185
*P. ucsense*	2HH T	OM914583	ON024157	ON024158	ON024159
*P. uruguayense*	CBS 143247 T	LT904729	LT904699	LT904698	MN969200
*P. vanderhammenii*	CBS 126216 T	GU981574	GU981647	KF296382	KF296458
*P. vasconiae*	CBS 339.79 T	GU981599	GU981653	KF296386	KF296459
*P. vickeryae*	BRIP72552a T	OP903479	OP921966	n.a.	OP921965
*P. viridissimum*	CGMCC 3.18796 T	KY495004	KY495113	KY494944	KY495059
*P. wotroi*	CBS 118171 T	GU981591	GU981637	KF296369	KF296460
*P. yunnanense*	CGMCC 3.18794 T	KY494990	KY495099	KY494930	KY495048
*P. zhanjiangense*	CGMCC 3.25206 T	**ON563149**	**ON231294**	**ON470837**	n.a.
*P. zonatum*	CBS 992.72 T	GU981581	GU981651	KF296380	KF296461
sect. *Exilicaulis*					
*P. allaniae*	BRIP 74886a T	OP903475	OP921956	OP921954	OP921955
*P. alutaceum*	CBS 317.67 T	AF033454	KJ834430	KP016768	JN121489
*P. amapaense*	URM 8639 T	OL764382	OL782590	OL782584	ON854925
*P. aotearoae*	PDD 107543 T	KT887874	KT887835	KT887796	MN969174
*P. arabicum*	CBS 414.69 T	KC411758	KP016750	KP016770	KP064574
*P. archerae*	BRIP 72549c T	OP903477	OP921961	n.a.	OP921960
*P. atrolazulinum*	CBS 139136 T	JX140913	JX141077	JX157416	KP064575
*P. atrosanguineum*	CBS 380.75 T	JN617706	KJ834435	KP016771	JN406557
*P. burgense*	CBS 325.89 T	KC411736	KJ834437	KP016772	JN406572
*P. canis*	NRRL 62798 T	KJ511291	KF900167	KF900177	KF900196
*P. catenatum*	CBS 352.67 T	KC411754	KJ834438	KP016774	JN121504
*P. chalabudae*	CBS 219.66 T	KP016811	KP016748	KP016767	KP064572
*P. cinerascens*	NRRL 748 T	AF033455	JX141041	JX157405	MN969112
*P. cinereoatrum*	CBS 222.66 T	KC411700	KJ834442	KP125335	JN406608
*P. citreonigrum*	CBS 258.29 T	AF033456	EF198621	EF198628	JN121474
*P. citreosulfuratum*	IMI 92228 T	KP016814	KP016753	KP016777	KP064615
*P. consobrinum*	CBS 139144 T	JX140888	JX141135	JX157453	KP064619
*P. corylophilum*	CBS 312.48 T	AF033450	JX141042	KP016780	KP064631
*P. cravenianum*	CBS 139138 T	JX140900	JX141076	JX157418	KP064636
*P. danzhouense*	CGMCC 3.25204 T	**ON563150**	**ON231295**	**ON470838**	n.a.
*P. decumbens*	CBS 230.81 T	AY157490	KJ834446	KP016782	JN406601
*P. diabolicalicense*	CBS 140967 T	KT887840	KT887801	KT887762	MN969175
*P. dimorphosporum*	CBS 456.70 T	AF081804	KJ834448	KP016783	JN121517
*P. dravuni*	F01V25 T	AY494856	n.a.	n.a.	n.a.
*P. erubescens*	CBS 318.67 T	AF033464	HQ646566	EU427281	JN121490
*P. fagi*	CBS 689.77 T	AF481124	KJ834449	KP016784	JN406540
*P. fundyense*	CBS 140980 T	KT887853	KT887814	KT887775	MN969176
*P. guttulosum*	NRRL 907 T	HQ646592	HQ646576	HQ646587	MG386247
*P. hemitrachum*	CBS 139134 T	FJ231003	JX141048	JX157526	KP064642
*P. hermansii*	CBS 124296 T	MG333472	MG386214	MG386229	MG386242
*P. heteromorphum*	CBS 226.89 T	KC411702	KJ834455	KP016786	JN406605
*P. katangense*	CBS 247.67 T	AF033458	KP016757	KP016788	KP064646
*P. kurssanovii*	CBS 625.67 T	EF422849	KP016758	KP016789	KP064647
*P. labradorum*	CBS 145775 T	MK881918	MK887898	MK887899	MK887900
*P. laeve*	CBS 136665 T	KF667369	KF667365	KF667367	KF667371
*P. lapidosum*	CBS 343.48 T	MN431392	KJ834465	FJ530984	JN121500
*P. maclennaniae*	CBS 198.81 T	KC411689	KJ834468	KP016791	KP064648
*P. melinii*	CBS 218.30 T	AF033449	KJ834471	KP016792	JN406613
*P. menonorum*	NRRL 50410 T	HQ646591	HQ646573	HQ646584	KF900194
*P. meridianum*	CBS 314.67 T	AF033451	KJ834472	KP016794	JN406576
*P. momoii*	CBS 139157 T	JX140895	JX141073	JX157479	KP064673
*P. namyslowskii*	CBS 353.48 T	AF033463	JX141067	KP016795	JF417430
*P. nepalense*	CBS 203.84 T	KC411692	KJ834474	KP016796	JN121453
*P. ovatum*	CBS 136664 T	KF667370	KF667366	KF667368	KF667372
*P. pagulum*	CBS 139166 T	JX140898	JX141070	JX157519	KP064655
*P. parvofructum*	CBS 141690 T	LT559091	LT627645	LT627646	MN969197
*P. parvum*	CBS 359.48 T	AF033460	HQ646568	KF900173	JN406559
*P. philippinense*	CBS 623.72 T	KC411770	KJ834482	KP016799	JN406543
*P. pimiteouiense*	CBS 102479 T	AF037431	HQ646569	HQ646580	JN406650
*P. punicae*	JMRC:SF:12421 T	n.a.	KX839673	KX839671	KX839675
*P. raciborskii*	CBS 224.28 T	AF033447	JX141069	KP016800	JN406607
*P. repensicola*	CBS 139160 T	JX140893	JX141150	JX157490	KP064660
*P. restrictum*	CBS 367.48 T	AF033457	KJ834486	KP016803	JN121506
*P. rubefaciens*	CBS 145.83 T	KC411677	KJ834487	KP016804	JN406627
*P. rubidurum*	CBS 609.73 T	AF033462	HQ646574	HQ646585	JN406545
*P. smithii*	CBS 276.83 T	KC411723	KJ834492	KP016806	JN406589
*P. striatisporum*	CBS 705.68 T	AF038938	MN969401	KP016807	JN406538
*P. subturcoseum*	CBS 139132 T	FJ231006	JX141161	JX157532	KP064674
*P. tenue*	CGMCC 3.25205 T	**ON563151**	**ON231296**	**ON470839**	**ON470842**
*P. terrenum*	CBS 313.67 T	AM992111	KJ834496	KP016808	JN406577
*P. velutinum*	CBS 250.32 T	AF033448	JX141170	MT478037	KP064682
*P. vinaceum*	CBS 389.48 T	AF033461	HQ646575	HQ646586	JN406555
*P. xanthomelinii*	CBS 139163 T	JX140921	JX141120	JX157495	KP064683
sect. *Talaromyces*					
*T. aculeatus*	CBS 289.48 T	KF741995	KF741929	KF741975	MH793099
*T. adpressus*	CGMCC 3.18211 T	KU866657	KU866844	KU866741	KU867001
*T. alveolaris*	CBS 142379 T	LT558969	LT559086	LT795596	LT795597
*T. amazonensis*	CBS 140373 T	KX011509	KX011490	KX011502	MN969186
*T. amestolkiae*	CBS 132696 T	JX315660	JX315623	KF741937	JX315698
*T. angelicae*	KACC 46611 T	KF183638	KF183640	KJ885259	KX961275
*T. annesophieae*	CBS 142939 T	MF574592	MF590098	MF590104	MN969199
*T. apiculatus*	CBS 312.59 T	JN899375	KF741916	KF741950	KM023287
*T. argentinensis*	NRRL 28750 T	MH793045	MH792917	MH792981	MH793108
*T. aspriconidius*	CBS 141835 T	MN864274	MN863343	MN863320	MN863332
*T. atkinsoniae*	BRIP 72528a T	OP059084	OP087524	n.a.	OP087523
*T. aurantiacus*	CBS 314.59 T	JN899380	KF741917	KF741951	KX961285
*T. aureolinus*	CGMCC 3.15865 T	MK837953	MK837937	MK837945	MK837961
*T. australis*	CBS 137102 T	KF741991	KF741922	KF741971	KX961284
*T. bannicus*	CGMCC 3.15862 T	MK837955	MK837939	MK837947	MK837963
*T. beijingensis*	CGMCC 3.18200 T	KU866649	KU866837	KU866733	KU866993
*T. brevis*	CBS 141833 T	MN864269	MN863338	MN863315	MN863328
*T. calidicanius*	CBS 112002 T	JN899319	HQ156944	KF741934	KM023311
*T. californicus*	NRRL 58168 T	MH793056	MH792928	MH792992	MH793119
*T. cnidii*	KACC 46617 T	KF183639	KF183641	KJ885266	KM023299
*T. coprophilus*	CBS 142756 T	LT899794	LT898319	LT899776	LT899812
*T. cucurbitiradicus*	ACCC 39155 T	KY053254	KY053228	KY053246	n.a.
*T. derxii*	CBS 412.89 T	JN899327	JX494306	KF741959	KM023282
*T. dimorphus*	CGMCC 3.15692 T	KY007095	KY007111	KY007103	KY112593
*T. domesticus*	NRRL 58121 T	MH793055	MH792927	MH792991	MH793118
*T. duclauxii*	CBS 322.48 T	JN899342	JX091384	KF741955	JN121491
*T. euchlorocarpius*	CBM PF1203 T	AB176617	KJ865733	KJ885271	KM023303
*T. flavovirens*	CBS 102801 T	JN899392	JX091376	KF741933	KX961283
*T. flavus*	CBS 310.38 T	JN899360	JX494302	KF741949	JF417426
*T. francoae*	CBS 113134 T	KX011510	KX011489	KX011501	MN969188
*T. funiculosus*	CBS 272.86 T	JN899377	MN969408	KF741945	KM023293
*T. fuscoviridis*	CBS 193.69 T	KF741979	KF741912	KF741942	MN969156
*T. fusiformis*	CGMCC 3.18210 T	KU866656	KU866843	KU866740	KU867000
*T. galapagensis*	CBS 751.74 T	JN899358	JX091388	KF741966	KX961280
*T. ginkgonis*	10725 T	OL638158	OL689844	OL689846	OL689848
*T. haitouensis*	HR1-7	MZ045695	MZ054634	MZ054637	MZ054631
*T. indigoticus*	CBS 100534 T	JN899331	JX494308	KF741931	KX961278
*T. intermedius*	CBS 152.65 T	JN899332	JX091387	KJ885290	KX961282
*T. kabodanensis*	CBS 139564 T	KP851981	KP851986	KP851995	MN969190
*T. kendrickii*	CBS 136666 T	KF741987	KF741921	KF741967	MN969158
*T. lentulus*	CGMCC 3.15689 T	KY007088	KY007104	KY007096	KY112586
*T. liani*	CBS 225.66 T	JN899395	JX091380	KJ885257	KX961277
*T. louisianensis*	NRRL 35823 T	MH793052	MH792924	MH792988	MH793115
*T. macrosporus*	CBS 317.63 T	JN899333	JX091382	KF741952	KM023292
*T. mae*	CGMCC 3.15690 T	KY007090	KY007106	KY007098	KY112588
*T. malicola*	NRRL 3724 T	MH909513	MH909406	MH909459	MH909567
*T. mangshanicus*	CGMCC 3.18013 T	KX447531	KX447530	KX447528	KX447527
*T. marneffei*	CBS 388.87 T	JN899344	JX091389	KF741958	KM023283
*T. muroii*	CBS 756.96 T	MN431394	KJ865727	KJ885274	KX961276
*T. mycothecae*	CBS 142494 T	MF278326	LT855561	LT855564	LT855567
*T. nanjingensis*	CCTCCM 2012167 T	MW130720	MW147759	MW147760	MW147762
*T. neofusisporus*	CGMCC 3.15415 T	KP765385	KP765381	KP765383	MN969165
*T. oumae-annae*	CBS 138208 T	KJ775720	KJ775213	KJ775425	KX961281
*T. panamensis*	CBS 128.89 T	JN899362	HQ156948	KF741936	KM023284
*T. penicillioides*	CGMCC 3.15822 T	MK837956	MK837940	MK837948	MK837964
*T. pinophilus*	CBS 631.66 T	JN899382	JX091381	KF741964	KM023291
*T. pratensis*	NRRL 62170 T	MH793075	MH792948	MH793012	MH793139
*T. primulinus*	CBS 321.48 T	JN899317	JX494305	KF741954	KM023294
*T. pseudofuniculosus*	CBS 143041 T	LT899796	LT898323	LT899778	LT899814
*T. purgamentorum*	CBS 113145 T	KX011504	KX011487	KX011500	MN969189
*T. purpureogenus*	CBS 286.36 T	JN899372	JX315639	KF741947	JX315709
*T. qii*	CGMCC 3.15414 T	KP765384	KP765380	KP765382	MN969164
*T. rapidus*	CBS 142382 T	LT558970	LT559087	LT795600	LT795601
*T. rosarhiza*	GUCC 190040.1 T	MZ221603	MZ333143	MZ333137	MZ333141
*T. ruber*	CBS 132704 T	JX315662	JX315629	KF741938	JX315700
*T. rubicundus*	CBS 342.59 T	JN899384	JX494309	KF741956	KM023296
*T. rufus*	CBS 141834 T	MN864272	MN863341	MN863318	MN863331
*T. santanderensis*	HF05 T	OP082331	OP067657	OP067656	OP067655
*T. sayulitensis*	CBS 138204 T	KJ775713	KJ775206	KJ775422	MN969146
*T. shilinensis*	XCW_SN259 T	OL638159	OL689845	OL689847	OL689849
*T. siamensis*	CBS 475.88 T	JN899385	JX091379	KF741960	KM023279
*T. soli*	NRRL 62165 T	MH793074	MH792947	MH793011	MH793138
*T. sparsus*	CGMCC 3.16003 T	MT077182	MT083924	MT083925	MT083926
*T. stellenboschiensis*	CBS 135665 T	JX091471	JX091605	JX140683	MN969157
*T. stipitatus*	CBS 375.48 T	JN899348	KM111288	KF741957	KM023280
*T. stollii*	CBS 408.93 T	JX315674	JX315633	JX315646	JX315712
*T. striatoconidium*	CBS 550.89 T	MN431418	MN969441	MN969360	MT156347
*T. thailandensis*	CBS 133147 T	JX898041	JX494294	KF741940	KM023307
*T. tumuli*	NRRL 62151 T	MH793071	MH792944	MH793008	MH793135
*T. veerkampii*	CBS 500.78 T	KF741984	KF741918	KF741961	KX961279
*T. verruculosus*	NRRL 1050 T	KF741994	KF741928	KF741944	KM023306
*T. versatilis*	IMI 134755 T	MN431395	MN969412	MN969319	MN969161
*T. virens*	CGMCC 3.25207 T	**ON563150**	**ON231297**	**ON470840**	**ON470841**
*T. viridis*	CBS 114.72 T	AF285782	JX494310	KF741935	JN121430
*T. viridulus*	CBS 252.87 T	JN899314	JX091385	KF741943	JF417422
*T. wushanicus*	CS17.05 T	MZ356356	MZ361347	MZ361354	MZ361361
*T. xishaensis*	CGMCC 3.17995 T	KU644580	KU644581	KU644582	MZ361364
*T. yunnanensis*	KUMCC 18-0208 T	MT152339	MT161683	MT178251	n.a.
*T. zhenhaiensis*	ZH3-18 T	MZ045697	MZ054636	MZ054639	MZ054633
Outgroup					
*A. glaucus*	CBS 516.65 T	EF652052	EF651887	EF651989	EF651934
*Hamigera avellanea*	CBS 295.48T	AF454075	EU021664	EU021682	EU021627
*P. expansum*	CBS 325.48T	AY373912	AY674400	DQ911134	JF417427
*P. glabrum*	CBS 125543 T	GU981567	GU981619	KM089152	JF417447
*T. chongqingensis*	CS26-67 T	MZ358001	MZ361343	MZ361350	MZ361357
*T. trachyspermus*	CBS 373.48 T	JN899354	KF114803	KJ885281	JF417432

Note: Numbers in boldface indicate newly submitted sequences. T means type strain.

## Data Availability

The names of the new species were formally registered in the database Fungal Names (https://nmdc.cn/fungalnames (accessed on 10 July 2023)). Specimens were deposited in the Herbarium Mycologicum Academiae Sinicae (https://nmdc.cn/fungarium/ (accessed on 18 February 2023)). Cultures were deposited in the China General Microbiological Culture Collection Center (https://cgmcc.net/ (accessed on 18 July 2023)). The newly generated sequences were deposited in GenBank (https://www.ncbi.nlm.nih.gov/genbank (accessed on 20 May 2022)).

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
