# Peer review of "Seven New Species of Eurotiales (Ascomycota) Isolated from Tidal Flat Sediments in China"

_jof, 2023, doi:10.3390/jof9100960_

Round 1

Reviewer 1 Report

This manuscript "Seven new species of Eurotiales (Ascomycota) isolated from tidal flat sediments in China" is an interesting piece of work worth publishing in JOF but it needs revisions before it is accepted. 

The comments and suggestions are annotated in the manuscript. 

The suthors should run PHI test for the new species. 

This manuscript "Seven new species of Eurotiales (Ascomycota) isolated from tidal flat sediments in China" is an interesting piece of work worth publishing in JOF but it needs revisions before it is accepted. 

The comments and suggestions are annotated in the manuscript. 

Author Response

We express our deep thanks to the reviewer for the valuable suggestions and detailed corrections including language. As you will see from the revised copy, we accepted almost all of these suggestions.

 The authors should run PHT test for the new species.

Accepted.

It is advised to use at least 2 strains of the new species in the phylogeny.

Unfortunately, the strains of these new species did not isolated again. Further field surveys are needed for gathering more cultures of these species.

Reviewer 2 Report

The authors are reporting new fungal taxa from tidal flats. I do think the manuscript is well written and interesting; however, in order to publish new species, the authors must compare their species to any other species within the genera and determine a difference. This difference cannot be solely based on molecular data.

Major issue: From what I saw, the authors downloaded all the sequences from GenBank, added their new sequences and showed that there was some molecular difference, and also indicated that there was a morphological difference between the sister taxa (based on phylogenetic analysis) and their new species. However, the authors must also compare their new species against any other species within the genus that does not have molecular data as they are not included in the initial phylogenetic analysis. The authors have not satisfied this criterion (at least in the manuscript) so the new species cannot be accepted according to the code. 

If the authors did this step and it was omitted in this version, they should add it and include a table with all the species that were left out of the phylogenetic analysis and show the morphological difference that distinguishes the new species from each of these.

The manuscript reads well.

Author Response

We express our deep thanks to the reviewer for the valuable suggestions. As you will see from the revised copy, we mainly accepted these suggestions.

REVIEWER 2 COMMENTS

From what I saw, the authors downloaded all the sequences from GenBank, added their new sequences and showed that there was some molecular difference, and also indicated that there was a morphological difference between the sister taxa (based on phylogenetic analysis) and their new species. However, the authors must also compare their new species against any other species within the genus that does not have molecular data as they are not included in the initial phylogenetic analysis. The authors have not satisfied this criterion (at least in the manuscript) so the new species cannot be accepted according to the code. If the authors did this step and it was omitted in this version, they should add it and include a table with all the species that were left out of the phylogenetic analysis and show the morphological difference that distinguishes the new species from each of these.

Partially accepted. Due to the sophisticated classification, the inter-specific relationships within each section (or series) of genera Aspergillus, Penicillium and Talaromyces were commonly used (Samson et al. 2014; Visagie et al. 2014; Yilmaz et al. 2014; Houbraken et al. 2020). In the present study, the newly described species along with all members of each section (or series) were used for phylogenetic analyses. Moreover, morphological distinctions and sequence divergences between the new taxa and their close relatives are discussed. Only P. dravuni was not included in the phylogenetic analyses in section Exilicaulis due to no sequence data are available for this species. However, the distinctions between P. dravuni and two newly described species P. danzhouense and P. tenue are compared in detail in the discussion.

Round 2

Reviewer 2 Report

The authors have added the necessary information to satisfy the taxonomic code, Thank you.